# Caudatin Isolated from *Cynanchum auriculatum* Inhibits Breast Cancer Stem Cell Formation via a GR/YAP Signaling

**DOI:** 10.3390/biom10060925

**Published:** 2020-06-18

**Authors:** Xing Zhen, Hack Sun Choi, Ji-Hyang Kim, Su-Lim Kim, Ren Liu, Yu-Chan Ko, Bong-Sik Yun, Dong-Sun Lee

**Affiliations:** 1Interdisciplinary Graduate Program in Advanced Convergence Technology and Science, Jeju National University, Jeju 63243, Korea; zhenxing19932013@gmail.com (X.Z.); seogwi12@naver.com (J.-H.K.); ksl1101@naver.com (S.-L.K.); liuren0308@gmail.com (R.L.); koyuchan94@gmail.com (Y.-C.K.); 2Subtropical/Tropical Organism Gene Bank, Jeju National University, Jeju 63243, Korea; choix074@jejunu.ac.kr; 3Division of Biotechnology, College of Environmental and Bioresource Sciences, Jeonbuk National University, Gobong-ro 79, Iksan 54596, Korea; bsyun@jbnu.ac.kr; 4Practical Translational Research Center, Jeju National University, Jeju 63243, Korea; 5Faculty of Biotechnology, College of Applied Life Sciences, Jeju National University, SARI, Jeju 63243, Korea

**Keywords:** cancer stem cells, mammospheres, caudatin, glucocorticoid receptor (GR), YAP

## Abstract

In the complex tumor microenvironment, cancer stem cells (CSCs), a rare population of cells, are responsible for malignant tumor initiation, metastasis, drug resistance and recurrence. Controlling breast CSCs (BCSCs) using natural compounds is a novel potential therapeutic strategy for clinical cancer treatment. In this study, a mammosphere assay-guided isolation protocol including silica gel, a C18 column, gel filtration, and high-pressure liquid chromatography was used to isolate an inhibitory compound from *Cynanchum auriculatum* extracts. The isolated inhibitory compound was identified as caudatin. Caudatin inhibited breast cancer cell proliferation, mammosphere formation and tumor growth. Caudatin decreased the CD44^+^/CD24^−^ and aldehyde dehydrogenase^+^ cell proportions and the levels of c-Myc, Oct4, Sox2, and CD44. Caudatin induced ubiquitin (Ub)-dependent glucocorticoid receptor (GR) degradation and blocked subsequent Yes-associated protein (YAP) nuclear accumulation and target gene transcription signals in BCSCs. These results show that the GR/YAP signaling pathway regulates BCSC formation and that caudatin may be a potential chemopreventive agent that targets breast cancer cells and CSCs.

## 1. Introduction

*Cynanchum auriculatum* Royle ex Wight is an Asclepiadaceae medicinal plant that is widely distributed in Asian countries, especially in China, Japan, India and Korea [1]. In the Chinese and Korean pharmacopeia, the tuberous root of *C. auriculatum* named “Baishouwu” or “Baekshuoh” has been used as a traditional herbal medicine to treat gastric ulcers, neurasthenia, edema and nephritis [2,3]. Pharmacological and phytochemical studies have reported that *C. auriculatum* has high application and economic value and displays beneficial antiviral [4], gastroprotective [5], antioxidant [6], antidepressant [7] and antitumor activities [8]. Caudatin is a type of C-21 steroid mainly isolated from *C. auriculatum*, which has been shown to have effective activity against cancers of different origins [9,10]. It has been reported that caudatin induces cancer cell apoptosis through C/EBP Homologous Protein (CHOP)-, p38-mitogen-activated protein kinase (MAPK)-, and c-Jun N-terminal kinase (JNK)-mediated upregulation of death receptor 5 expression [11]. Furthermore, caudatin suppresses uterine cancer cells by regulating α-induced protein 1 (TNFAIP1)/Nuclear factor-κB (NF-κB) signaling [12]. Although evidence of the anticancer effects of caudatin is expanding, there is no report concerning the mechanisms underlying the effects of caudatin on breast cancer stem cells (BCSCs).

Breast cancer, one of the most lethal adenocarcinomas, is defined as uncontrolled breast cell growth; this cancer originates in the ducts or lobules of the breast and metastasizes to other regions of the human body, and it is a major cause of female fatality worldwide [13,14]. Triple-negative breast cancer (TNBC) remains the most challenging subtype of breast cancer to treat and is found in 15–20% of all breast cancer patients [15,16]. Therapeutic strategies including surgery, chemotherapy, radiotherapy, hormone therapy, and personalized targeted therapy with a combination therapy are often applied in clinical practice [17]. In recent decades, BCSCs with the metabolic marker aldehyde dehydrogenases (ALDH) and marker CD44, a small subpopulation of breast cancer cells, have been found to display stem cell capacities, such as self-renewal, long-term repopulation and differentiation [18]. In breast cancer therapy, BCSCs are responsible for cancer initiation, recurrence and metastasis [19]. A rare BCSC population shows multidrug resistance and radioresistance properties [20]. Therefore, novel strategies targeting BCSCs may be effective clinical therapies for breast cancer patients.

During the differentiation and metabolism of the mammary gland, steroid hormones regulate a series of biological processes through progesterone receptor (PR), estrogen receptor (ER) and glucocorticoid receptor (GR) interactions [21]. Nevertheless, GR activation is also linked with breast cancer heterogeneity and metastasis [22]. It has been reported that glucocorticoid (GC), a steroid hormone superfamily, promotes breast cancer cell survival and inhibits chemosensitivity [23]. GR influences microRNA (miRNA) profiles and is responsible for the increased mortality of TNBC patients [24]. GR signaling activates the Hippo pathway to promote breast cancer progression, BCSC self-renewal and chemoresistance [25,26].

The Hippo pathway is composed of a large protein network and is highly conserved in mammals, controlling cell growth, tissue regeneration, and tumorigenesis [27]. The main molecules of the Hippo pathway include related serine/threonine kinases, mammalian STE20-like protein kinase 1/2 (MST1/2), and large tumor suppressor 1/2 (LATS1/2) [28]. LATS1/2 phosphorylates Yes-associated protein (YAP) and the coactivator PDZ-binding motif (TAZ) to induce the degradation of YAP and TAZ. The transcription factor complex of YAP/TAZ and transcriptional enhanced associate domains (TEAD) regulates cell growth and survival gene expression [29]. Aberrant activation of YAP/TAZ occurs via cancer stem cell (CSC) traits, epithelial-mesenchymal transition (EMT), treatment resistance and metastasis [30]. Novel drug development focused on targeting the YAP signaling pathway may expand the limited therapies available for the treatment of breast cancer.

In this study, anti-CSC compounds from *C. auriculatum* were isolated and tested. The purified compound caudatin inhibited BCSC formation. We show that caudatin has anti-CSC activity via a GR/YAP signaling pathway.

## 2. Materials and Methods

### 2.1. Chemical and Reagents

Silica gel 60 and thin layer chromatography (TLC) plates were purchased from MERK (Darmstadt, Germany), and Sephadex LH-20 was obtained from Pharmacia (Uppsala, Sweden). High-pressure liquid chromatography (HPLC) was performed with a Shimadzu application system (Shimadzu, Kyoto, Japan). Cell viability was measured using the EZ-Cytox Cell Viability Assay Kit (DoGenBio, Seoul, Korea). Caudatin (Purity is >98%) was obtained from ChemFaces Co. (Hubei, China). Caudatin were stored at −20 °C as 100 mM dissolved in 100% dimethyl sulfoxide (DMSO). The final DMSO concentration used was <0.1% and control group was treated with DMSO only.

### 2.2. Plant Material

Samples of *C. auriculatum* were purchased from verified market sources (Daegu, Korea). The samples (No. 2018_03) are deposited in the Department of Biomaterial, Jeju National University, Jeju Si, South Korea.

### 2.3. Extraction and Isolation

Ground samples of *C. auriculatum* were extracted with methanol. The isolation method is summarized in Figure 1A. The powder created from *C. auriculatum* was solubilized with 7.5 L of methanol. The methanol extracts were concentrated and mixed with water, and the methanol fraction was evaporated. The water-suspended fraction was extracted with 1X ethyl acetate. The ethyl acetate-concentrated portion was concentrated, and the concentrated parts were loaded onto a silica gel column (3 × 30 cm) and fractionated with solvent (chloroform:methanol, 20:1) (Appendix A). The five portions were fractionated and tested by evaluating mammosphere formation. The #2 fraction potentially inhibited mammosphere formation. The #2 fraction was loaded onto a reversed-phase C18 column (ODS) open column (5 × 10 cm) and eluted in four fractions (Appendix A). The four fractions were assayed by evaluating mammosphere formation. The 30–50% fraction showed inhibition of mammosphere formation. The 30–50% fraction was loaded onto a Sephadex LH-20 column (2.5 × 30 cm) and eluted in three fractions (Appendix A). The three fractions were assayed by evaluating mammosphere formation. Fraction #1 showed inhibition of mammosphere formation. Fraction #1 was isolated using preparatory TLC (glass plate; 20 × 20, layer thickness; 210–270 µm) and developed in the chloroform-methanol solvent (20:1) using a TLC glass chamber for 2 h. After development, the plates were removed and dried, and the fluorescence was visualized under UV radiation (UV_254nm_ and UV_365nm_). Individual bands were separated from the silica gel plate and eluted with methanol. Each fraction was assayed by evaluating mammosphere formation (Appendix A). The #3 fraction was loaded onto a Shimadzu HPLC 20A instrument (Shimadzu, Tokyo, Japan). HPLC was performed with an ODS 10 × 250-mm C18 column (flow rate; 2 mL/min). The sample was prepared and sieved through a 0.2 µm syringe filter for HPLC analysis. The injection volume for HPLC isolation was 500 µL. The mobile phase was composed of water (solvent A) and acetonitrile (solvent B). For elution, the acetonitrile proportion was initially set at 20%, increased to 80% at 20 min and finally increased to 100% at 30 min (Appendix A). The purified sample was detected at a retention time of 35 min.

The #3 fraction was loaded onto a Shimadzu HPLC instrument (Shimadzu, Tokyo, Japan). HPLC used an ODS 10 × 250 mm column (flow rate; 2 mL/min). For elution, the acetonitrile proportion was initially set at 20%, increased to 80% at 20 min and finally increased to 100% at 30 min (Appendix A).

### 2.4. Structural Analysis

Electrospray ionization (ESI) mass spectrometry was conducted on a QTRAP-3200 mass spectrometer (Applied Biosystems, Foster City, CA, USA). Nuclear magnetic resonance (NMR) spectra were obtained on a JEOL JNM-ECA 600 fourier transform-nuclear magnetic resonance (FT-NMR) spectrometer at 600 MHz for ^1^H NMR and at 150 MHz for ^13^C NMR in CD_3_OD. The molecular structure of the isolated compound was determined by mass spectrometry and NMR. Its molecular weight was identified to be 490 Da by ESI mass spectrometry, which showed ion peaks at *m/z* 513.0 [M+Na]^+^ in the positive mode and at *m/z* 489.0 [M−H]^−^ in the negative mode. The ^1^H NMR spectrum in CDCl_3_ exhibited signals due to two olefinic methine protons at δ 5.46 and 5.31, two oxygenated methine protons at δ 4.50 and 3.53, two methine and seven methylene protons at δ 1.05–2.81, and six methyl protons at δ 2.12, 2.07, 1.36, 1.08, 1.01, and 1.01. In the ^13^C NMR spectrum, the 28 carbon peaks detected included two carbonyl carbons at δ 208.8 and 165.8; two sp^2^ quaternary carbons at δ 166.7 and 140.4; two sp^2^ methine carbons at δ 117.6 and 112.8; three oxygenated quaternary carbons at δ 91.4, 88.0, and 74.1; two oxygenated methine carbons at δ 71.4 and 70.6; two quaternary carbons at δ 57.7 and 37.0; two methine carbons at δ 43.6 and 38.0; seven methylene carbons at δ 38.6, 38.5, 34.1, 33.0, 31.7, 28.7, and 24.1; and six methyl carbons at δ 27.0, 20.8, 20.7, 18.4, 16.4, and 9.3 (Appendix A). All proton-bearing carbons were assigned by the heteronuclear multiple quantum coherence (HMQC) spectrum, and the ^1^H-^1^H COSY spectrum revealed six partial structures (Appendix A). Further structural elucidation was performed with the aid of the heteronuclear multiple bond correlation (HMBC) spectrum, which showed long-range correlations from the methyl proton at δ 1.08 to the carbons at δ 140.4, 43.6, 38.6, and 37.0; from the methyl proton at δ 1.36 to the carbons at δ 91.4, 88.0, 71.4, and 57.7; from the methyl proton at δ 2.07 to the carbons at δ 166.7, 112.8, and 38.0; from the methine proton at δ 5.31 to the carbons at δ 74.1 and 37.0; from the methine protons at δ 5.46 and 4.50 to the ester carbonyl carbon at δ 165.8; and from the methylene protons at δ 2.81/1.77 and the methyl proton at δ 2.12 to the ketone carbonyl carbon at δ 208.8 (Appendix A). Therefore, the molecular structure of the isolated compound was identified as that of caudatin (Figure 2). Other HMBC correlations were well matched to caudatin.

### 2.5. Cell Culture and Mammosphere Formation Assay

MDA-MB-231 and MCF-7 human breast cancer cells were obtained from the American Type Culture Collection (Rockville, MD, USA) and maintained in Dulbecco’s modified Eagle’s medium (DMEM) supplemented with 10% (V/V) fetal bovine serum (FBS; HyClone, Thermo Fisher Scientific, CA, USA) and 1% penicillin/streptomycin (Gibco, Thermo Fisher Scientific, CA, USA) in a 5% CO_2_ incubator. Breast cancer cells were incubated at 1 × 10^4^ cells per well in an ultralow-attachment 6-well plate with MammoCult^™^ culture medium (STEMCELL Technologies, Vancouver, BC, Canada) supplemented with hydrocortisone (0.48 μg/mL), heparin (4 μg/mL), and caudatin (100 μM) for 7 days. To establish secondary mammosphere, primary mammosphere was collected into individual conical tubes by gentle centrifugation at 1000 rpm for 1 min. After removal of supernatant, cell pellet was trypsinized with 1X ethylenediaminetetraacetic acid (EDTA)/Trypsin, followed by replating a single cell suspension at a density of 1 × 10^4^ cells/well in ultra-low attachment 6-well plates containing 2 mL of complete MammoCult™ medium. In 5 days, the number and size of the mammosphere were assessed compared with control. Mammosphere formation was quantified using the NIST’s integrated colony enumerator (NICE) program [31]. Mammosphere formation was determined by examining the mammosphere formation efficiency (MFE) (%). MFE (%) were measured using the formula (number of sphere with control or drug/number of sphere with control (DMSO) x100) [32].

### 2.6. Cell Proliferation

Breast cancer cells were seeded at 1.5 × 10^4^ cells per well in a 96-well plate for 24 h and incubated with caudatin at several concentrations (25, 50, 100, 150, 300 and 400 µM) for 24 h. Then, proliferation was assayed using an EZ-Cytox kit (DoGenBio, Seoul, Korea) in accordance with the manufacturer’s protocol, and the OD_450_ was measured using a VERSA max microplate reader (Molecular Devices, San Jose, CA, USA).

### 2.7. Annexin V/Propidium Iodide (PI) Assay to Detect Cell Apoptosis and Hoechst 33,342 Staining of Apoptotic Nuclei

MDA-MB-231 cells were cultured in 6-well plates with caudatin (100 µM) for 24 h. Apoptosis was assayed using an Annexin V/PI staining kit according to the manufacturer’s instructions (BD, San Jose, CA, USA). The stained cells were analyzed with an Accuri C6 (BD, San Jose, CA, USA). MDA-MB-231 cells were incubated with 100 µM caudatin for 24 h, and then the cells were stained with a Hoechst 33,342 staining solution (Invitrogen™, Thermo Fisher Scientific, CA, USA) for 10 min at 37 °C in accordance with the manufacturer’s protocol. The cells were observed using the CELENA^®^ S Digital Imaging System (Logos Biosystems, Anyang, Korea).

### 2.8. Scratch Assay

MDA-MB-231 cancer cells were seeded in a 6-well plate at 2 × 10^6^ cells/well with DMEM/10% FBS. A scratch was made by using a microtip after the cells had grown into a monolayer. After washing two times with 1× PBS, the cancer cells were cultured with caudatin at a concentration of 50 or 100 µM in fresh DMEM/0.5% FBS for 12 h. Photomicrographs of the wounded areas were acquired using a light microscope. The cells that migrated across the white lines were counted in five randomly chosen fields from each triplicate treatment. The percentage of inhibition was expressed using untreated wells at 100%.

### 2.9. Transwell Assay

We followed a previously described method [33]. Invasion and migration assays were performed using 8-µm-pore polycarbonate membranes (Merck Millipore, Darmstadt, Germany) coated with/without a Matrigel matrix basement (BD, San Jose, CA, USA) in 24-well hanging inserts. Two hundred microliters of MDA-MB-231 cell suspensions (1 × 10^5^ cells) treated with 100 µM caudatin in DMEM supplemented with 0.5% FBS were added to the upper chamber. Then, 900 µL of DMEM containing 20% FBS as a chemoattractant was added to the bottom chamber. The cancer cells were maintained at 37 °C in a 5% CO_2_ incubator for two days. The cells that passed through the membrane were fixed with 4% paraformaldehyde and stained with 0.03% crystal violet. Images were acquired using an inverted light microscope.

### 2.10. Colony Formation Assay

MDA-MB-231 cells (5 × 10^2^ cells/well) were cultured in a 6-well plate, incubated with caudatin at various concentrations (50, 100, 150, 300 and 400 µM) in DMEM and incubated for 7 days. The colonies were fixed using 3.7% formaldehyde for 10 min and stained with 0.05% crystal violet for 30 min. The colonies that grew were imaged by using a scanner (Umax PowerLook 1100, lasersoft Imaging, Seoul, Korea).

### 2.11. Flow Cytometric Analysis and ALDH1 Activity

We used a previously described method [34]. After caudatin (100 µM) treatment for 24 h, MDA-MB-231 cells were harvested using 1× trypsin/EDTA. The samples were incubated with anti-CD24 PE-conjugated and anti-CD44 FITC-conjugated antibodies (BD, San Jose, CA, USA) on ice for 20 min. The cancer cells were washed three times with fluorescence activated cell sorter (FACS) buffer and analyzed on an Accuri C6 cytometer (BD, San Jose, CA, USA). The ALDH activity of MDA-MB-231 cells was examined using an ALDEFUOR^™^ assay kit (STEMCELL Technologies, Vancouver, BC, Canada). We used a previously described method [34]. The cancer cells were treated with caudatin (100 µM) for 24 h and incubated in ALDH assay buffer at 37 °C for 30 min. The ALDH-positive cells were assayed with an Accuri C6 (BD, San Jose, CA, USA).

### 2.12. Western Blot Analysis

Protein samples were extracted from mammospheres and cancer cells using lysis buffer. After electrophoresis using 10% sodium dodecyl sulphate-polyacrylamide gel electrophoresis (SDS-PAGE), the proteins were transferred to a PVDF membrane (Millipore, Burlington, MA, USA). The membrane was blocked with an Odyssey blocking buffer for 1 h at room temperature and then incubated overnight with primary antibodies. The antibodies were anti-c-Myc (551101, BD, San Jose, CA, USA); anti-Oct4 (LF-MA30482) and anti-GAPDH (LF-PA0018) (AbFrontier, Seoul, Korea); anti-YAP (sc-101199), anti-Lamin B (sc-365962), anti-CD44 (sc-7297), and anti-Sox2 (sc-365923) (Santa Cruz Biotechnology, Dallas, TX, USA); and anti-GR (#12041, Cell Signaling Technology, Danvers, MA, USA). After membranes were washed three times using Tris-buffered saline/Tween 20, all membranes were incubated with IRDye 680RD- and IRDye 800W-labeled secondary antibodies for 1 h at room temperature, and the signal images were determined with an Odyssey CLx (Li-Cor, Lincoln, NE, USA). Densitometric analysis of western blot has been done using Image Studio Ver 5.2 program of Odyssey CLx (Li-Cor, Lincoln, NE, USA).

### 2.13. Immunoprecipitation (IP)

Protein samples were extracted from mammospheres using IP lysis buffer containing a protease inhibitor. After the detection of protein concentrations, all samples (450 µg per sample) were incubated with 30 µL of Puredown Protein A/G-Agarose (GenDEPOT, DAWINbio, Hanam, Korea) for clearing of nonspecific proteins. According to the manufacturer’s protocol, an anti-GR monoclonal antibody (mAb) (#12041, Cell Signaling Technology, Danvers, MA, USA) was incubated with the samples overnight at 4 °C. Then, 50 µL of Puredown Protein A/G-Agarose was mixed with each sample and maintained with gentle rotation at 4 °C for 4 h. After centrifugation and washing with IP lysis buffer, all samples were analyzed by western blot analysis.

### 2.14. Gene Expression Analysis

We used a previously described method [35]. Isolated RNA from MDA-MB-231 cells and mammospheres was extracted and purified, and real-time RT-quantitative PCR was performed using a one-step RT-qPCR kit (Enzynomics, Daejeon, Korea). The specific primers are described in Table 1.

### 2.15. Immunofluorescence (IF) Assay

After caudatin (100 µM) treatment for 24 h, MDA-MB-231 cells were fixed with 3.7% paraformaldehyde for 20 min, permeabilized with 0.5% Triton X-100 for 15 min, blocked with 3% bovine serum albumin (BSA) for 60 min, stained with a rabbit anti-GR mAb (#12041, Cell Signaling Technology, Danvers, MA, USA) and mouse anti-YAP mAb (sc-101199, Santa Cruz Biotechnology, Dallas, TX, USA), followed by staining with an anti-rabbit Alexa 488-conjugated secondary antibody (A32731) and anti-mouse Alexa 555-conjugated secondary antibody (A32727) (ThermoFisher, Waltham, MA, USA). Finally, the nuclei of cancer cells were stained with 4′, 6-diamidino-2-phenylindole (DAPI), and GR and YAP were visualized with a fluorescence microscope (Lionheart, Biotek, VT, USA).

### 2.16. Small Interfering RNA (siRNA)

We transfected MDA-MB-231 cells with human GR- and YAP-specific siRNAs (Bioneer, Daejeon, Korea) to determine the effects of the GR and YAP proteins on mammosphere formation. For siRNA transfection, cancer cells were incubated with Lipofectamine 2000 (Invitrogen, Carlsbad, CA, USA) according to the manufacturer’s protocol. The levels of the GR and YAP proteins were examined by western blot analysis.

### 2.17. Xenograft Transplantation

The nude mouse experiments were performed as described previously [36]. Female four-week-old nude mice obtained from OrientBio (Seoul, South Korea) were kept in independent-ventilation cages with access to food and water for one week. Fourteen female nude mice injected with MDA-MB-231 cells (2 × 10^6^ cells/mouse) were divided into a negative control group and a caudatin (10 mg/kg) treatment group. Mouse tumor volumes were measured and calculated for one month using the formula (width^2^ × length)/2. Animal care and experiments were performed in accordance with protocols approved by the Institutional Animal Care and Use Committee (IACUC) of Jeju National University.

### 2.18. Statistical Analysis

All data were analyzed with GraphPad Prism 7.0 software (GraphPad Prism, Inc., San Diego, CA, USA). All data from three independent experiments are reported as the mean ± standard deviation (SD). Data were analyzed using one-way ANOVA. A *p*-value less than 0.05 was considered significant.

## 3. Results

### 3.1. BCSC Inhibitor from C. auriculatum

To isolate a human BCSC inhibitor, a mammosphere formation assay was performed with MDA-MB-231 cells treated with extracts from *C. auriculatum*. The mammosphere formation assay-guided BCSC inhibitor isolation method is summarized in Figure 1A. An isolated sample derived from *C. auriculatum* methanol extracts inhibited BCSCs (Figure 1B). Methanol/ethyl acetate extraction, silica gel, an open C-18 column, gel filtration, preparatory TLC, and HPLC were used for compound isolation and purification. The compound isolated with the mammosphere assay-guided protocol was confirmed by using HPLC (Figure 1C). The molecular structure of the purified compound was determined using mass and NMR data, and the molecule was identified as caudatin (Figure 2).

### 3.2. Caudatin Suppresses Cell Growth and Mammosphere Formation

We assessed the inhibitory effect of caudatin on breast cancer cells. Caudatin showed an antiproliferative effect on MDA-MB-231 cells at several concentrations. After 24 h of stimulation, we observed an antiproliferative effect of caudatin (≤50 µM) that occurred in a dose-dependent manner (Figure 3A). To evaluate whether caudatin can suppress mammosphere formation, various doses of caudatin were applied to mammospheres derived from MDA-MB-231 and MCF-7 (Appendix A) cells. Caudatin not only inhibited the number but also decreased the size of the mammospheres (Figure 3B). In order to compare caudatin activity, we used a ciclesonide as mammosphere formation inhibitor (Figure 3B). Caudatin inhibits secondary mammosphere derived from MCF-7 and MDA-MB-231 cell (Appendix A). Compared with control treatment, treatment with caudatin for 24 h induced cell apoptosis (Figure 3C). Annexin V-FITC and PI staining showed an increased proportion of early apoptotic cells in MDA-MB-231 cells (Figure 3C). Hoechst staining data showed that the formation of apoptotic bodies was induced by caudatin (Figure 3D). Treatment with caudatin suppressed the colony formation, migration and invasion of breast cancer cells (Figure 3E–G). Our data showed that caudatin suppressed breast cancer cell growth and mammosphere formation, as well as colony formation, migration and invasion.

### 3.3. Caudatin Suppresses Xenograft Tumor Growth

As caudatin showed significant anticancer effects in vitro, in vivo studies were performed to evaluate the effects of this compound on tumor growth in more depth. There was no significant body weight difference between control and caudatin-treated mice (Figure 4A). At each time point, the tumor volume (Figure 4B) and weight (Figure 4C) of the caudatin-treated mice were smaller than those of the untreated mice. Our data showed that caudatin effectively reduced tumor growth.

### 3.4. Caudatin Decreases the Populations of CD44^+^/CD24^−^ and ALDH-expressing Cancer Cells

The BCSC population is characterized by the CD44^+^/CD24^−^ phenotype and ALDH1 expression to maintain malignancy [37,38]. The effect of caudatin treatment on the population of CD44^+^/CD24^−^ breast cancer cells was assayed. Caudatin decreased the CD44^+^/CD24^−^ cell fraction from 75.8% to 68.6% (Figure 5A). After caudatin treatment, the ALDH-positive cancer cell subpopulation was also evaluated to investigate this inhibitory effect. Caudatin decreased the fraction of ALDH-positive cancer cells from 14.4% to 1.4% (Figure 5B). Our data indicated that caudatin specifically inhibited the expression of BCSC markers.

### 3.5. Caudatin Blocks the GR Signal through the Ubiquitin (Ub)-Dependent Degradation of GR in BCSCs

To investigate the molecular mechanism of caudatin activity against BCSCs, we determined the level of GR in MDA-MB-231 mammospheres. We found that the total GR level was significantly reduced after caudatin treatment (Figure 6A). There was a compensatory increase in the GR gene transcript level under caudatin treatment (Figure 6B). After caudatin treatment, we performed IP for GR and western blotting with an anti-ubiquitin antibody using mammosphere samples. The level of ubiquitinated GR was increased (Figure 6C). MG132, a proteasome inhibitor, suppressed the GR degradation induced by caudatin, suggesting that caudatin increased Ub-related GR degradation. Caudatin-induced downregulation of GR was significantly antagonized by MG-132 (Figure 6D). Furthermore, the cytosolic and nuclear levels of the GR protein in mammospheres (Figure 6E) and IF levels of GR in MDA-MB-231 cells (Figure 6F) indicated that the levels of cytosolic and nuclear GR were reduced in caudatin-treated cells. To evaluate the role of the GR protein in BCSCs, we examined mammosphere formation with GR-specific siRNA. MDA-MB-231 cells treated with the GR-specific siRNA showed a 50% reduction in tumorsphere formation ability (Figure 6G). RU486, a GR antagonist, inhibited mammosphere formation in breast cancer cells (Figure 6H). The data suggested that GR signaling regulated mammosphere formation and that caudatin induced a malfunction in GR signaling through Ub-dependent degradation in BCSCs.

### 3.6. Caudatin Regulates the GR-YAP Signaling Pathway

We tested whether caudatin inhibits the GR-YAP signaling pathway to regulate the formation of BCSCs. Our results showed that the YAP total protein level (Figure 7A) and transcript level (Figure 7B) were not changed under caudatin treatment. The protein level in nuclear extracts from mammospheres (Figure 7D) and IF levels of YAP in MDA-MB-231 cells (Figure 7C) showed that the expression levels of nuclear YAP were significantly reduced in caudatin-treated cells. GR-knockdown MDA-MB-231 cells treated with the GR-specific siRNA showed a reduction in YAP protein expression (Figure 7E). MDA-MB-231 cells treated with YAP-specific siRNA showed a 50% reduction in tumorsphere formation ability (Figure 7F). Verteporfin, an inhibitor of the YAP-TEAD interaction, dramatically inhibited mammosphere formation in breast cancer cells (Figure 7G). YAP target genes (CTGF and CYR61) were examined in mammospheres. Our results showed that the transcript levels of the CTGF and CYR61 genes were decreased under caudatin, GR-specific siRNA or verteporfin treatment (Figure 7H–J). These data indicated that caudatin regulated the GR-YAP signal needed for BCSC formation.

### 3.7. Caudatin Regulates the Expression Levels of CSC Marker Proteins and Mammosphere Growth

To test whether caudatin regulates CSC marker gene expression, we assessed the protein expression of marker genes by western blot analysis. Caudatin treatment significantly decreased the expression of stem cell markers, including c-Myc, Oct4, Sox2 and CD44, in mammospheres derived from MDA-MB-231 cells (Figure 8A). To confirm that caudatin inhibited mammospheres, we treated mammospheres with caudatin and quantified the number of breast cancer cells derived from the mammospheres. Caudatin increased the death of cancer cells and decreased the cell number in the mammospheres (Figure 8B). Our data suggested that caudatin inhibited the growth of BCSCs by blocking the GR/YAP signaling pathway (Figure 8C).

## 4. Discussion

Breast cancer is a frequently reported malignant adenocarcinoma that develops in breast tissue and causes high morbidity and mortality among women worldwide [39]. Although treatments including surgery, chemotherapy and radiotherapy are performed to eradicate the primary tumor, breast cancer is still a fatal disease in many patients [40]. TNBC is one of the most clinically challenging breast cancer subtypes and is characterized by high risks of metastasis and recurrence and a low 5-year survival rate [41]. In clinical studies of TNBC and the tumor microenvironment, Breast CSCs, a subpopulation of tumor cells characterized by high ALDH expression and a CD44^high^/CD24^low^ phenotype, show multiple chemoresistance mechanisms [42]. Increasing evidence has shown that targeting BCSCs has high therapeutic efficacy [43,44].

Current studies have explained the anticancer activity of *C. auriculatum* extracts and the potential mechanisms involved [45,46]. However, caudatin, a C-21 steroid derived from *C. auriculatum*, has not been investigated in selective cytotoxicity against human CSCs. We isolated the active components from *C. auriculatum* extracts that possessed anti-CSC activity against mammospheres derived from breast cancer cells by activity-guided fractionation (Figure 1). The isolated compound was identified as caudatin (Figure 2). In a variety of cancer cells, caudatin suppresses proliferation and induces caspase-dependent apoptosis [12,47]. Our data showed that caudatin inhibited the proliferation of breast cancer cells and the growth of mammospheres. An apoptosis was induced in breast cancer cells by caudatin treatment (Figure 3). The metastasis of breast cancer is attributed to BCSC existence and is known to induce high recurrence and mortality in female patients [48]. Caudatin inhibited malignant behaviors of breast cancer, including migration, invasion, and colony formation (Figure 3). Furthermore, caudatin inhibited tumor growth in a xenograft mouse model (Figure 4).

Chemotherapeutics targeting BCSCs are capable of inhibiting the initiation of new tumors and cancer relapse. In current studies, three subtypes of BCSCs including mesenchymal-like (M) CD44^+^/CD24^−^ BCSCs, epithelial-like (E) ALDH^+^ BCSCs, and E/M ALDH^+^/CD44^+^/CD24^−^ BCSCs have been identified [49]. CD44, a class I transmembrane glycoprotein, is associated with regulating mesenchymal-like steps, such as cell migration, invasion, and adhesion [50]. ALDH catalyzes aldehyde oxidation and converts aldehydes into carboxylic acids, which is related to chemotherapeutic agent resistance in cancer cells [42,51]. Caudatin decreased the subpopulation of CD44^high^/CD24^low^ positive cells and ALDH-expressing populations in breast cancer cells (Figure 5). 

In the treatment of hematological malignancies and solid tumors, natural and synthetic GCs are widely used. GCs play a number of essential physiological roles, such as controlling metabolic homeostasis [52] and regulating the immune response [53] and cardiovascular function [54]. However, serious side effects, including the promotion of tumor growth and cancer metastasis, were occurred with GC clinical therapeutics [55]. Due to their lipophilic nature, GCs bind to GR, a specific receptor protein encoded by the NR3C1 gene that is expressed in almost all human cells, and GC-GR complex translocates into nucleus and regulates transcription of target genes [56]. Clinical evidence showed that the activation of GR induces chemoresistance and a poor prognosis in breast cancer patients [57]. The mammosphere formation was suppressed by GR knockdown with a GR-specific siRNA (Figure 6). Selective GR antagonists and modulators that have structures similar to those of GCs interact with GR and result in the inhibition of downstream gene transcription [58]. RU-486, a GR antagonist, inhibited mammosphere formation. Caudatin blocked the GR signal through Ub-dependent GR degradation in BCSCs (Figure 6). Our results show that GR signaling is important for BCSC survival.

In breast cancer, stimulation of GR induces Hippo pathway dysregulation and transcriptional coactivator YAP activation, nuclear accumulation and transcriptional activity [25]. Aberrant activation of YAP/TAZ leads to tumorigenesis in several tissues and confers CSC traits that induce EMT, metastasis, and drug resistance [28]. CTGF and CYR61, as downstream effectors of YAP/TAZ transcription, improve cell proliferation and migration [59]. YAP knockdown with YAP-specific siRNA suppressed mammosphere formation. Verteporfin, a YAP inhibitor that disrupts the YAP-TEAD interaction [60], inhibited BCSC formation and YAP target gene transcription. We showed that the YAP protein plays a vital role in BCSC maintenance. Caudatin reduced YAP nuclear translocation and transcriptional activity. Caudatin and GR-specific siRNA induced decreases in CTGF and CYR61 transcription. Caudatin inhibited BCSCs through the GR/YAP signaling pathway (Figure 7 and Figure 8). Caudatin may be an anticancer agent that controls breast cancer cells and BCSCs.

## 5. Conclusions

In this study, a compound isolated from *C. auriculatum* extracts was identified as caudatin by mass spectrometry and NMR. Our data showed that caudatin suppressed the growth of breast cancer cells, mammosphere formation and tumor growth in a nude mouse model. Caudatin decreased the sizes of the CD44^+^/CD24^−^ and ALDH1^+^ cell populations and the protein levels of c-Myc, Oct4, Sox2, and CD44. Caudatin treatment induced Ub-dependent GR degradation and blocked subsequent YAP nuclear accumulation and target gene (CTGF and CYR61) transcription signals in BCSCs. These results suggest that the GR/YAP signaling pathway regulates BCSC formation and that caudatin may be a potential anticancer agent that targets breast cancer cells and BCSCs.

## Figures and Tables

**Figure 1 biomolecules-10-00925-f001:**
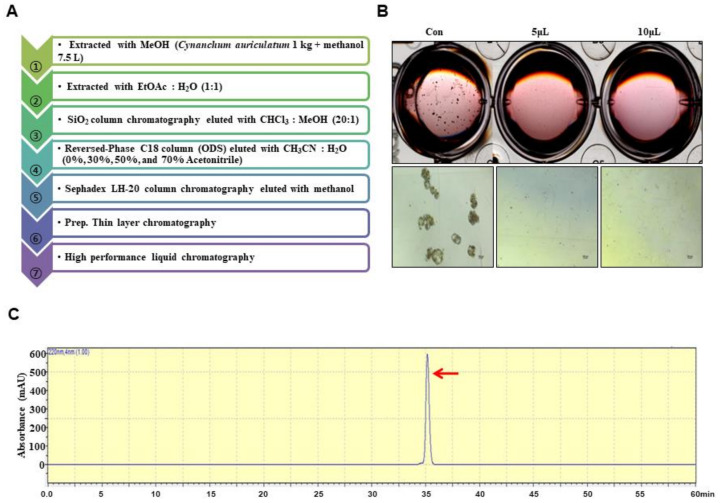
Purification of a cancer stem cell (CSC) inhibitor derived from *Cynanchum auriculatum* and assessment of extracts using a mammosphere formation assay. (**A**) Isolation procedure for the mammosphere inhibitor. (**B**) Inhibition of mammospheres using high performance liquid chromatography (HPLC)-purified samples. MDA-MB-231 cells were treated with HPLC extracts or dimethyl sulfoxide (DMSO) in CSC culture medium for 7 days. Images show representative mammospheres and were obtained by microscopy (scale bar: 100 μm). (**C**) HPLC chromatogram of the inhibitor isolated from *Cynanchum auriculatum*.

**Figure 2 biomolecules-10-00925-f002:**
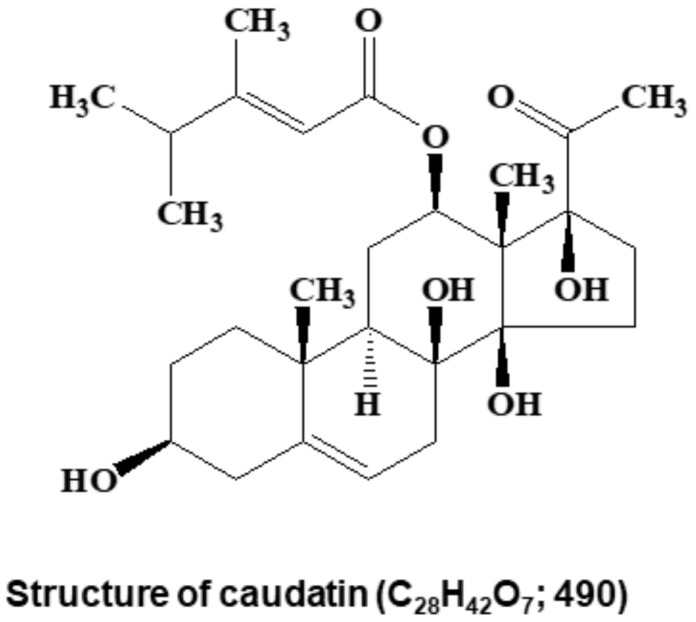
Chemical structure of the CSC inhibitor caudatin isolated from *Cynanchum auriculatum*.

**Figure 3 biomolecules-10-00925-f003:**
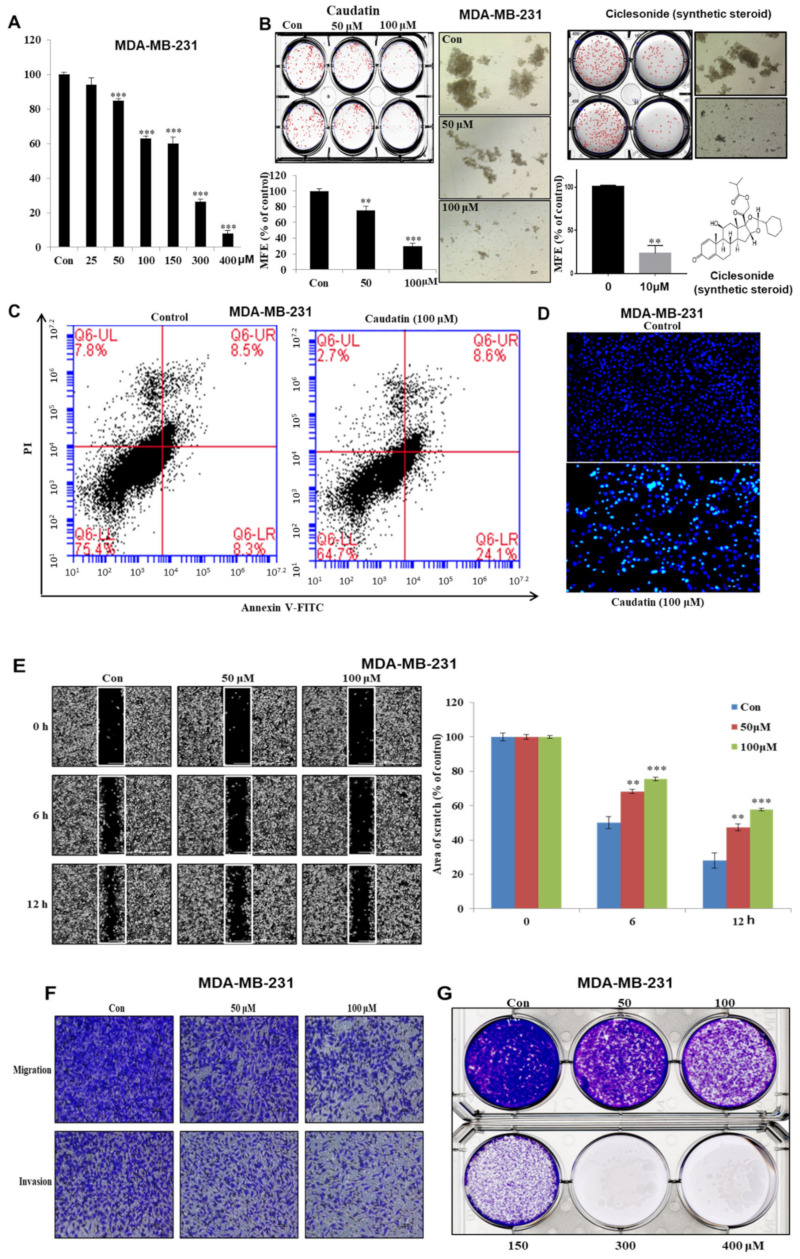
The effects of caudatin on the proliferation and mammosphere formation of breast cancer cells. (**A**) MDA-MB-231 cells were cultured in a 96-well plate with caudatin for 24 h. MDA-MB-231 cell proliferation was assayed with an EZ-Cytox kit. (**B**) Caudatin and ciclesonide reduced mammosphere formation by breast cancer cells. Mammospheres derived from MDA-MB-231 cells were cultured in ultralow-attachment 6-well plates with CSC culture medium for seven days. The mammosphere formation efficiency (MFE) was examined with increasing concentrations of caudatin and ciclesonide. Images show representative mammospheres and were obtained by microscopy (scale bar: 100 μm). (**C**) Caudatin induced apoptosis in MDA-MB-231 cells. Apoptosis was determined using Annexin V/propidium iodide (PI) staining. (**D**) Apoptotic cells induced by caudatin were stained with Hoechst 33,342 dye (scale bar: 50 μm). (**E**) The migration of MDA-MB-231 cells with/without caudatin (Dulbecco’s Modified Eagle’s medium (DMEM)/0.5% fetal bovine serum (FBS)) was imaged at 0, 6 h and 12 h by a scratch assay (scale bar: 100 μm). The percent of inhibition in cell migration was expressed using untreated well at 100%. (**F**) The cell migration (without Matrigel) and invasion (with Matrigel) of MDA-MB-231 cells exposed to caudatin were determined by transwell assays (scale bar: 100 μm). (**G**) MDA-MB-231 cells were incubated in 6-well plates and treated with caudatin. Representative colony formation data were collected. The data from triplicate experiments are represented as the mean ± SD. ** *p* < 0.01 and *** *p* < 0.001 versus the DMSO-treated control group.

**Figure 4 biomolecules-10-00925-f004:**
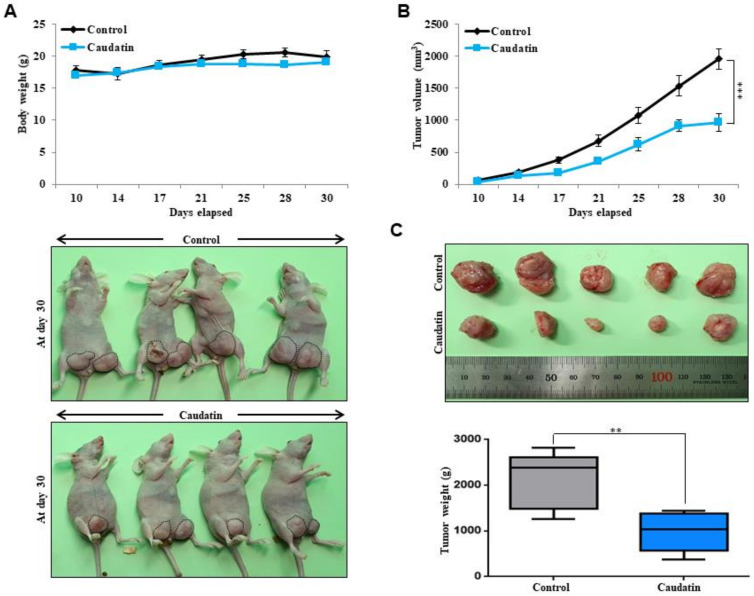
The effect of caudatin on tumor growth in a mouse model. MDA-MB-231 cells (2 × 10^6^ cells/mouse) were injected into the mammary fat pad of female nude mice and treated with caudatin (10 mg/kg) or DMSO (control group, n = 6). (**A**) The body weights of the mice in the caudatin-treated group were comparable to those of the mice in the control group. On day 30, images of the mice in the control and caudatin-treated groups were captured with a camera. (**B**) Tumor volume was measured using a caliper and calculated as (width^2^ × length)/2 at the indicated time points. (**C**) On day 30, the tumor weights of the control and caudatin-treated mice were assayed after sacrifice. The data are presented as the mean ± SD of three independent experiments. ** *p* < 0.01 and *** *p* < 0.001 versus the DMSO-treated control group.

**Figure 5 biomolecules-10-00925-f005:**
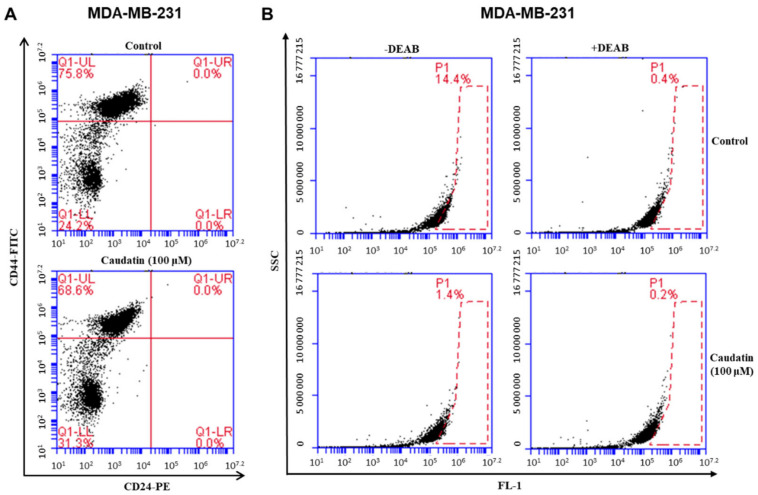
The effects of caudatin on the CD44^high^/CD24^low^ and aldehyde dehydrogenase (ALDH)-positive cell proportions in a breast cancer cell line. (**A**) The CD44^high^/CD24^low^ cell population of MDA-MB-231 cells treated with caudatin (100 μM) or DMSO for 24 h was analyzed by flow cytometry. The gating was based on the binding of a control antibody (red cross). (**B**) MDA-MB-231 cells were treated with caudatin (100 μM) for 24 h and subjected to an ALDEFLUOR assay and FACS analysis. Representative flow cytometric data are shown. The right panel shows the ALDH-positive population treated with the ALDH inhibitor DEAB, and the left panel represents the ALDH-positive population without N,N-diethylaminobenzaldehyde (DEAB) treatment.

**Figure 6 biomolecules-10-00925-f006:**
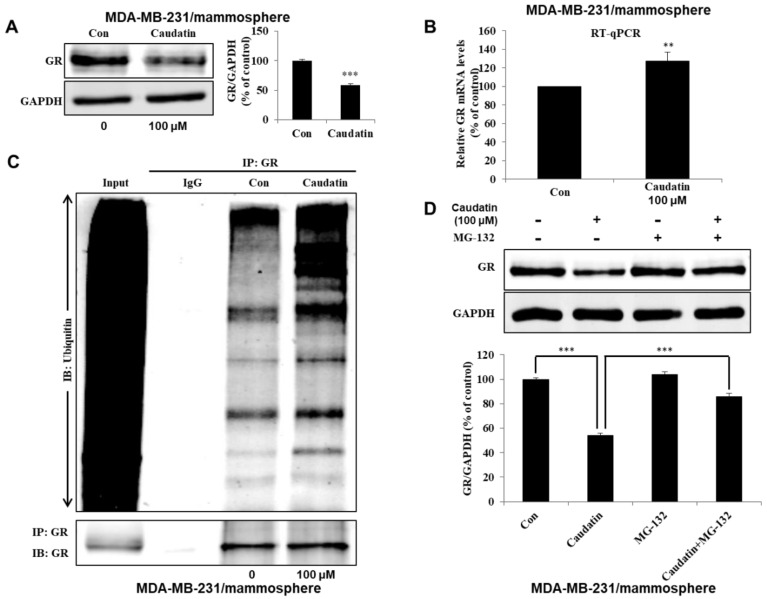
Caudatin inhibits the glucocorticoid receptor (GR) signal through the ubiquitin (Ub)-dependent degradation of GR. (**A**) The expression level of GR in the total protein of MDA-MB-231-derived mammospheres was measured after treatment of the mammospheres with caudatin for 48 h using western blot analyses. (**B**) After treatment with caudatin, the transcriptional expression of the GR gene was detected in mammospheres by real-time reverse transcription- quantitative polymerase chain reaction (RT-qPCR) using specific primers. β-actin was used as an internal control. (**C**) Mammospheres incubated with caudatin were lysed for immunoprecipitation (IP) with an anti-GR antibody, and western blot analysis was performed using anti-ubiquitin and anti-GR antibodies. (**D**) Mammospheres were incubated with caudatin and MG-132 (0.5 μM) for 24 h and lysed for western blot analysis. (**E**) The expression level of GR in the cytosolic and nuclear protein fractions of MDA-MB-231-derived mammospheres was measured after treatment of the mammospheres with caudatin using western blot analysis. (**F**) Immunofluorescence (IF) analysis of GR (green) expression and localization in MDA-MB-231 cells under caudatin treatment was performed. (**G**) The effect of knocking down GR expression using GR-specific siRNA on mammosphere formation was evaluated. (**H**) The effect of RU-486, an antagonist of GR, on mammosphere formation was evaluated. The data are presented as the mean ± SD of three independent experiments. ** *p* < 0.01 and *** *p* < 0.001 versus the DMSO-treated control group.

**Figure 7 biomolecules-10-00925-f007:**
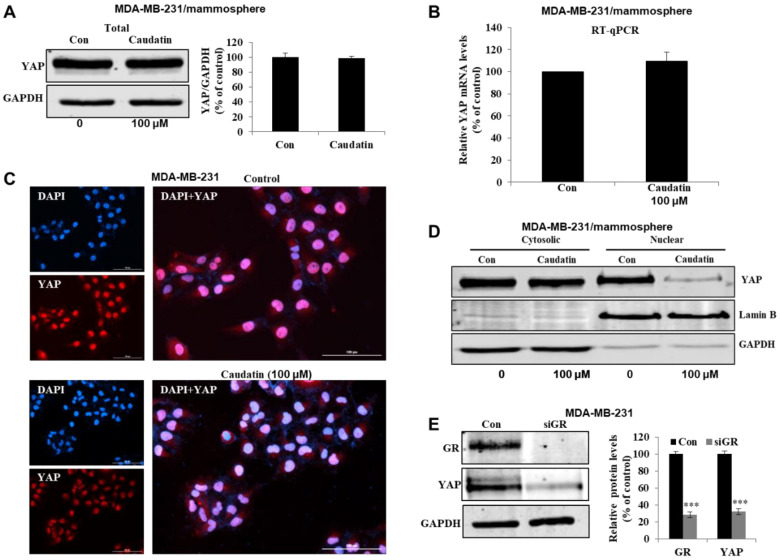
Caudatin reduces YAP nuclear localization and regulates breast CSC formation. (**A**) The expression level of YAP in the total protein of MDA-MB-231-derived mammospheres was measured after treatment of the mammospheres with caudatin for 48 h using western blot analysis. (**B**) The transcriptional expression of the YAP gene was detected in mammospheres by real-time RT-qPCR using specific primers. β-actin was used as an internal control. (**C**) Immunofluorescence (IF) analysis of YAP (red) expression and localization in MDA-MB-231 cells under caudatin treatment was performed. (**D**) The expression level of YAP in the cytosolic and nuclear protein fractions of MDA-MB-231-derived mammospheres was measured after treatment of the mammospheres with caudatin using western blot analysis. (**E**) The expression levels of GR and YAP in MDA-MB-231 cells were measured after treatment with a siRNA specific for GR. (**F**) The effect of knocking down YAP expression using YAP-specific siRNA on mammosphere formation was evaluated. (**G**) The effect of verteporfin, a suppressor of the YAP/TEAD complex, on mammosphere formation was evaluated. (**H**–**J**) After treatment with caudatin, siRNA targeting GR or verteporfin, the transcriptional expression of the CYR61 and CTGF genes was detected by real-time RT-qPCR using specific primers. β-actin was used as an internal control. The data are presented as the mean ± SD of three independent experiments. ** *p* < 0.01 and *** *p* < 0.001 versus the DMSO-treated control group.

**Figure 8 biomolecules-10-00925-f008:**
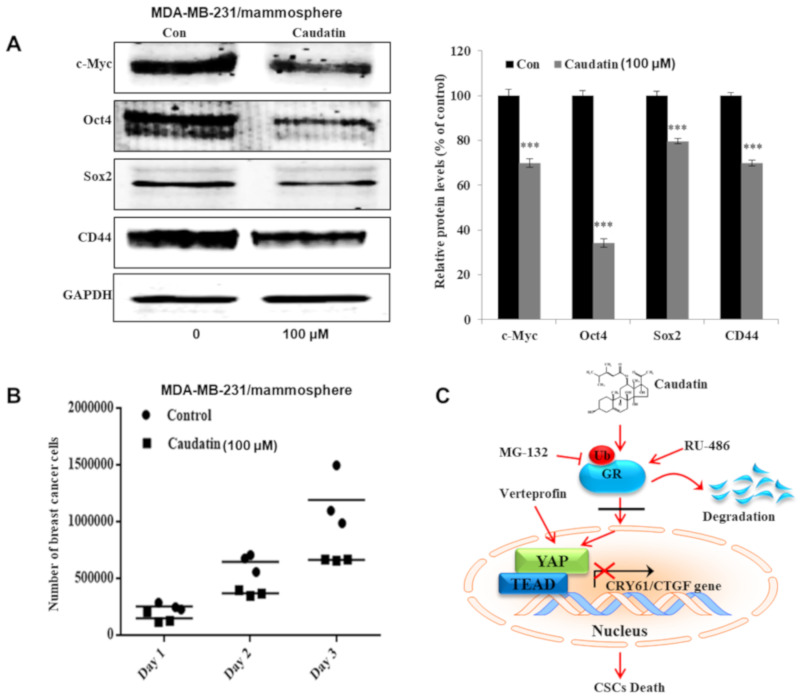
The effects of caudatin on the expression of CSC marker proteins and mammosphere growth. (**A**) Western blot analyses of the c-Myc, Oct4, Sox2 and CD44 proteins in mammospheres after treatment with caudatin. (**B**) Caudatin inhibits mammosphere growth. Mammospheres with/without caudatin treatment were dissociated into single cells and cultured in 6-cm dishes in equal numbers. One, two and three days later, the cells were counted. (**C**) The proposed model of breast CSC death mediated through the GR degradation and YAP signaling pathways induced by caudatin is shown. The data are presented as the mean ± SD of three independent experiments. *** *p* < 0.001 versus the DMSO-treated control group.

**Table 1 biomolecules-10-00925-t001:** Specific Real-time RT-qPCR primer sequences containing human *GR, YAP, CTGF, CYR61,* and *β-actin* genes.

Genes	Primers
GR	Forward: 5′-GAAGGAAACTCCAGCCAGAA-3′ Reverse: 5′-CAGCTAACATCTCGGGGAAT-3′
YAP	Forward: 5′- GAACCCCAGATGACTTCCTG-3′ Reverse: 5′-CTCCTTCCAGTGTTCCAAGG-3′
CTGF	Forward: 5′-CCAATGACAACGCCTCCTG-3′ Reverse: 5′-TGGTGCAGCCAGAAAGCTC-3′
CYR61	Forward: 5′-AGCCTCGCATCCTATACAACC-3′ Reverse: 5′-TTCTTTCACAAGGCGGCACTC-3′
β-actin	Forward: 5′-TGTTACCAACTGGGACGACA-3′ Reverse: 5′-GGGGTGTTGAAGGTCTCAAA-3′

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
