# Peer review of "Caudatin Isolated from Cynanchum auriculatum Inhibits Breast Cancer Stem Cell Formation via a GR/YAP Signaling"

_biomolecules, 2020, doi:10.3390/biom10060925_

Round 1
Reviewer 1 Report
The manuscript reports a novel anticancer activity of the steroid natural products isolated from the traditional herbal drug. Caudatin has been shown to inhibit growth and induction of cancer stem cells via glucocorticoid receptor associated pathway. As a steroid compound, caudatin may be then considered a promising lead or for further development as a potential drug against cancer resistance to treatments. The methodology is complex and well selected for the purpose. The results have been painstakingly described and a lot of data created. The topic and the well-performed experiments would justify publication in Biomolecules. Unfortunately, there are several minor and one major problem that must be dealt with before acceptance.
The minor issues are in the missing details of Material and Methods:
- The information about the HPLC instrument (model, setup)
- Purity of commercial caudatin standard
- Information about TLC preparative plates (SG60?, thickness?)
- TLC conditions (eluent, time)
- NMR and MS instrumentation details
- Cell culture medium – hydrocortisone and heparin concentrations
- How was caudatin dissolved and dosed to the cell cultures?
The major and very important problem is lack of any reference drug or positive control in ALL the experiments! Hence, it is very difficult to assess an actual strength of caudatin activities. If the Authors have such data, please provide them. Then, the Discussion seems to ignore this problem as well. There is no in-depth analysis of literature on other similar (or not) natural products that caudatin could be compared with. Please revise the Discussion accordingly.
Author Response
We attached comments of reviewer 1.

Reviewer 2 Report
In the article submitted for publication on biomelecules Zhen et al. deal with the effect of caudatin, isolated from cynanchum auriculatum, in reducing proliferation, mammosphere formation and tumor growth, using the triple negative breast cancer line mda-mb231 as an experimental model. From the results obtained the authors suggest that the caudatin mechanism of action is carried out through the regulation of the pathway of GR/YAP.
In my opinion the authors should:
-do not employed the generic term inhibitor in the abstract (line 23);
-In scratch assay: evaluate the percentage of wound closure, considering in the various conditions always the same point. Evaluate also a shorter time and specify the culture conditions used in materials and methods (in particular the percentage of serum used).
-colony formation assay: specify how the images shown in figure 3G were obtained. Were they obtained using an inverted light microscope as reported in methods?
-they should specify how they got the photos of the MW showing in the figures
-specify in which experimental model the results shown were obtained. For example, were the evaluation of ALDH1 activity and the flow cytometric analysis of CD44 / CD24 carried out in the mda-mb231? while the other evaluation in cells obtained from mammospheres? Why?
-They should be assessed in cells obtained from mammosfere the enrichment in stem cells than the MDA-MB231.
-they should use at least secondary mammospheres
- they should specify the compound (caudatin) used in the experiments in what it is diluted and use the right control for the experiments. Why in some experiments do they deal with control with DMSO?
-they should show a greater magnification in figure 3D to show the changes induced by the compound in the chromatin.
- The authors should report how the calculation for MFE was obtained (fig 3B).
- the authors should report less overexposed results in western blotting experiments, especially with regard to GADPH internal control. The differences are not often appreciated. They should also specify how densitometric analysis was achieved.
- The authors should report the right controls in the nucleus / cytosol localization experiments in order to highlight that there is no contamination in the fractions obtained (e.g. for the cytosol they should report lamin B).
-the authors, in figure 6D, should report the significance of the caudatin + MG132 sample compared to the only caudatin sample.
Author Response
We attached comments of reviewer 2.

Reviewer 3 Report
The paper by Zhen et al. reports that caudatin, a natural compound isolated from Cynanchum auriculatum, is able to inhibit breast cancer cell proliferation, induce cell apoptosis and suppress tumor growth in vivo. Most importantly, caudatin can target breast cancer stem cells by downregulation of glucocorticoid receptor and YAP signaling pathway. Overall, the data are interesting and support a novel anticancer mechanisms of caudatin. Thus, this paper deserves to be published. However, there are some points to clarify.
In the figure 3 (C-D) it is shown that an incubation of MDA-231 cells with100 microM caudatin for 24 hours induced cell apoptosis. Why this caudatin concentration and incubation time were chosen to evaluate the effect of this compound on cell migration, invasion, mammosphere and colony formation? It is difficoult to define the impact of the treatment on these specific cell behaviours considering that a significant percentage of cells is already dying. It should be better to perform these functional assays using sub-apoptotic concentrations.
In Figures 6-8, the concentration of caudatin used to perform the experiments are not indicated.
Moreover, in my opinion, the language style of the discussion might be improved.
Author Response
We attached comments of reviewer 3.

Round 2
Reviewer 1 Report
The revised manuscript seems to have most of the suggestion followed and the missing essential data have been provided. In my opinion, the manuscript can be accepted for publication in Biomolecules.
Reviewer 2 Report
After the revision the manuscript has improved.
Reviewer 3 Report
The revised version of the manuscript can be published on Biomolecules